# Sway and Acceleration Changes of the Center of Mass during Walking Stance Phase before and after Total Knee Arthroplasty

**DOI:** 10.3390/geriatrics8010002

**Published:** 2022-12-24

**Authors:** Takashi Fukaya, Hirotaka Mutsuzaki, Koichi Mori

**Affiliations:** 1Department of Physical Therapy, Faculty of Health Sciences, Tsukuba International University, 6-8-33 Manabe, Tsuchiura 300-0051, Japan; 2Centre for Medical Sciences, Ibaraki Prefectural University of Health Sciences, 4669-2 Ami, Ibaraki 300-0394, Japan; 3Department of Radiological Sciences, Ibaraki Prefectural University of Health Sciences, 4669-2 Ami, Ibaraki 300-0394, Japan

**Keywords:** knee osteoarthritis, center of mass, lateral sway, root mean square, acceleration change, total knee arthroplasty

## Abstract

Elucidating the sway and changes in the acceleration of center of mass (COM) during walking is important for effective gait training and rehabilitation. The objective of this study was to verify the improvement in gait before and after total knee arthroplasty (TKA) from COM sway and the changes in the acceleration of COM during the stance phase of walking. This study included 13 patients (1 male and 12 females) with medial knee osteoarthritis who were hospitalized for TKA. The COM sway during the stance phase of walking was evaluated using root mean square (RMS) normalized by walking speed, and the changes in acceleration were further verified. Lateral and vertical RMS showed significant differences between preoperative and postoperative states and demonstrated low values after TKA. The lateral acceleration at the latter part of the early stance phase demonstrated a significant difference between preoperative and postoperative states. A significant difference was also observed in the lateral acceleration in the late stance phase between the two groups. Improvement in pain and alignment after TKA reduced the lateral sway of COM and the changes in acceleration during the gait stance phase, which is speculated to lead to improvement in gait and prevention of falls.

## 1. Introduction

Knee osteoarthritis (KOA) is an age-related disease. It is characterized by degeneration of the joint cartilage that leads to structural and functional failure of the knee joint caused by formation of bone osteophytes, meniscal injury, muscle weakness, and a limited range of motion [1,2]. In the load joint, mechanical factors play an important role in the development and progression of KOA [3].

In walking with medial KOA, the external knee adduction moment due to the appearance of varus thrust associated with varus deformity is often reported to increase the mechanical load on the medial side of the knee joint [4,5]. Several gait modifications have the ability to alter knee load [6], and exercise and gait retraining programs can alter the external knee adduction moment in people with KOA [7]. The trunk is tilted toward the affected side or toe-out to reduce the load on the knee joint due to the external knee adduction moment associated with varus deformity [8,9].

The tilt of the trunk in patients with medial KOA has been reported to shift the center of mass (COM) to the stance side on the frontal plane during walking and acts to reduce the external knee adduction moment [10]. On the other hand, the tilt of the trunk affects the first peak of the external knee adduction moment, but the displacement of COM due to the tilt of the trunk does not [11].

Control of COM position and acceleration over the base of support during walking is an indicator of dynamic stability, and the lack of such control ability potentially leads to falls [12]. The ability to maintain walking stability and control lateral direction stability is diminished with age [13]. COM acceleration contributes to walking stability [14], and a decrease in mediolateral COM accelerations alters gait patterns in the elderly [15]. Thus, elucidating the sway and changes in the acceleration of COM during walking is important for the development of effective gait training and rehabilitation for physical therapists. However, only a few reports on the COM sway during walking in KOA are available [16].

As represented by varus thrust and increase in the external knee adduction moment, patients with end-stage KOA have an altered pattern of walking prior to total knee arthroplasty (TKA). TKA is the most common treatment for patients with severe KOA with advanced dysfunction. It has been reported to effectively improve pain and function in self-reported assessments, including the Western Ontario and McMaster Universities Osteoarthritis Index [17,18]. However, satisfactory results on walking ability have not yet been obtained postoperatively [19]. In addition, postoperative gait function and biomechanics are often measured several months after TKA surgery, and whether abnormal gait is the result of preoperatively adopted gait patterns or surgical intervention is not well understood [20]. Therefore, recognizing functional improvement and residual disability in the early postoperative period is considered helpful for early rehabilitation after TKA with the aim of recovering knee joint function and improving gait. There are few measurements of changes in center of gravity sway due to walking at a relatively early stage after TKA, and we believe that presenting the results is novel.

The objective of this study was to verify the improvement in gait before and after TKA from the COM sway and the changes in the acceleration of COM during the stance phase of walking. We hypothesized that TKA would reduce the lateral sway of COM and the changes in acceleration to improve gait by decreased pain and modified knee alignment.

## 2. Materials and Methods

### 2.1. Participants

This study included 13 patients (1 male and 12 females) with medial KOA who were hospitalized for TKA. All patients were admitted to our hospital, and gait analysis was performed preoperatively and at 5 weeks postoperatively. The severity of KOA was classified using the Kellgren–Lawrence scale [21]. An orthopedic surgeon measured the femorotibial angle on radiographs. Varus deformity was defined as a femorotibial angle > 180°. In addition, the numerical rating scale was used to assess knee pain, and the range of motion was measured. The preoperative and postoperative characteristics of the patients are presented in Table 1.

The exclusion criteria for this study were histories of rheumatoid arthritis, surgery on the knee joint, trauma, or central neuropathy. Patients who had difficulty walking on their own were also excluded.

All patients provided written informed consent for their participation in this study, and the study was approved by the ethics committee of our institution. This study was conducted in accordance with the principles of the Declaration of Helsinki.

### 2.2. TKA Surgery

For patients under general anesthesia, surgery was performed using a tourniquet. A medial parapatellar approach was used through a midline skin incision. An intramedullary alignment rod was used for femoral cutting, and an intramedullary or extramedullary guide system was used for tibial cutting. The NexGen or PERSONA CR implants (Zimmer, Warsaw, IN, USA) were used for the femoral component, and the NexGen Trabecular Metal Monoblock Tibia or NexGen CR Stem Tibia implants were used for the tibial component. The patella was not replaced, and the posterior cruciate ligament was retained. Following surgery, the patient underwent prophylactic intravenous antibiotic therapy for 3 days (1 g cefazolin every 12 h). A foot pump (Novamedix A-V Impulse System; Kobayashi Medical, Osaka, Japan) and anti-embolism stockings (AnsilkW; ALCARE, Tokyo, Japan) were used for thromboembolic prophylaxis [22,23].

### 2.3. Procedure

Kinematic data were obtained at 200 Hz using an eight-camera motion analysis system (Vicon Nexus, Vicon, Oxford, UK) to measure walking. The ground reaction force (GRF) data were recorded at 1200 Hz using two floor-mounted force plates (Kistler Instruments, Winterthur, Switzerland) to obtain kinetic data, and these data were synchronized with the motion capture data. The global coordinate system was defined with the X-axis as anterior–posterior, the Y-axis as lateral, and the Z-axis as vertical.

The patients walked barefoot along a level walkway at their normal speed, and an average of three gait trials was conducted for each patient. In this study, three-dimensional motion analysis with a lower extremity model of the Plug-In-Gait marker set [24] was used. Reflective markers of 9.5 mm diameter were affixed to the following anatomic landmarks: anterior and posterior superior iliac spines, lateral thighs, lateral femoral epicondyles, lateral shanks, lateral malleoli, calcanei, and dorsa of the feet at the base of the second metatarsal. After the reflective markers were attached, each patient was instructed to stand barefoot for a single static calibration before gait analysis. Patients were then instructed to step on a floor-mounted force plate using the lower limb and were allowed to perform several preparation trials. The procedure for these measurements was the same before and after TKA.

The following patient data were also recorded: height, weight, leg length (anterior superior iliac spine to medial malleolus), anterior superior iliac spine width, knee joint width, and ankle joint width. These data were obtained prior to the walking measurements.

### 2.4. Data Analysis

The marker and joint angle position data acquired from the Plug-In-Gait model and the GRF data were low-pass filtered using a Butterworth filter. Using GRF data, the section from the initial contact to toe-off was used as the walking stance phase for analysis. The midpoint was calculated from the left and right posterior superior iliac spine markers and used as COM of the patient, and the second-order differentiation of position of COM was performed to calculate the acceleration in the three-axis direction. The root mean square (RMS) was calculated by the following formula from the changes in the acceleration of COM during the walking stance phase.
$$\mathrm{RMS}=\sqrt{\frac{1}{n}\sum_{i=1}^{n}\alpha_{i}{}^{2}}$$

In the above formula, *α* indicates acceleration, and *n* indicates the number of frames of measurement data.

Since a larger RMS indicates that more impaired dynamic stability has been reported [25], the RMS value was used as an index of the dynamic stability in this study. The RMS of the acceleration of the center of gravity of the body is affected by walking speed. In this study, we confirmed the correlation coefficient between walking speed and RMS. The results show a significant correlation in some variables (Table 2), and thus it was adjusted by dividing the RMS value by the squared value of the walking speed according to past reports [25,26].

In addition, on the basis of the RMS results during the walking stance phase, the characteristic points of the curve were extracted for the changes in the COM acceleration in the vertical and lateral directions during the walking stance phase.

In this study, early stance showed data up to 20% of the stance phase, and late stance showed data after 80% of the stance phase.

### 2.5. Statistical Analysis

The differences between preoperative and postoperative RMS of the COM and the characteristic points of the curve of the COM acceleration in the vertical and lateral directions were verified using paired t-tests, as were the differences between alignment, pain, and range of motion. In addition, the effect size (d) was calculated for each variable. Statistical significance was set at *p* < 0.05. All statistical analyses were performed using IBM SPSS 23.0 Statistics (IBM Corp. Released 2015. IBM SPSS Statistics for Windows, Version 23.0. IBM Corp., Armonk, NY, USA).

## 3. Results

### 3.1. Walking Speed and Sway of the COM between Groups

In this study, there was no significant difference in walking speed between pre and post TKA (*p* = 0.22). Table 3 shows the RMS results during the walking stance phase. No significant difference was found in the anterior–posterior RMS between the two groups. Lateral and vertical RMS showed significant differences between the two groups and demonstrated low values after TKA.

### 3.2. Acceleration Changes in the Three Axes of COM between Groups

Figure 1 shows the change in COM acceleration during the walking stance phase. The lateral acceleration change did not show a significant difference in the medial acceleration at the first part of the early stance phase, but the lateral acceleration at the latter part of the early stance phase showed a significant difference between the two groups (Table 4). A significant difference was also observed in the lateral acceleration in the late stance phase between the two groups (Table 4). The vertical acceleration showed a significant difference in the upward acceleration that occurred in the late stance phase between the two groups (Table 5).

## 4. Discussion

The results of this study showed that after TKA, the lateral sway of COM in the walking stance phase was decreased, and the changes in the acceleration of COM in the early and late phases of stance were decreased. These two improvements support the hypothesis of this study.

KOA is a typical orthopedic disease in which a high rate of gait abnormalities is observed, and it involves various dysfunctions such as pain, range of motion limitation, and muscle weakness [27,28,29]. The lateral sway and the changes in the acceleration of COM in the lateral direction observed in the study results were large at the initial stance phase before the surgery. This result was observed at the same time as the varus thrust in the early phase of stance, which is frequently seen in severe medial knee osteoarthritis. The appearance of varus thrust due to structural changes in the knee joint causes lateral agitation of COM, and as a result, the external knee adduction moment increased and the mechanical load on the medial side of the knee joint increased.

In addition, in hip OA, abnormalities have been reported to be more likely to occur in the double limb support than in the single limb support, and the lateral sway of COM is greater in the double limb support [30]. The results of this study also showed that the magnitude of the acceleration change before and after TKA increased at the timing of double limb support in the early and late phases of stance, and the magnitude of the change tended to increase significantly before TKA. The early phase of stance is the time to absorb the effect from the GRF, and control of the knee joint is important. However, strategies such as misalignment due to varus deformity and lateral bending of the trunk to avoid pain have been reported before TKA [31]. As compensation, the results of this study suggest that the upward acceleration change in the late stance phase increased. Due to these factors, in medial KOA, the lateral acceleration change becomes large in the early and late phases of stance. After TKA, the lateral sway and acceleration change of COM decreases due to improvement in pain and alignment, and gait improvement was observed.

The ability to control medial–lateral stability and recover from falls decreases with age [32], and understanding lateral sway during walking in the medial–lateral direction with older persons is important for the development of effective gait training and rehabilitation [30].

Recently, analysis using smartphones and general cameras has been reported [33]. Although there are issues such as measurement accuracy that can withstand the validity of clinical measurement results, these methods are clinically simple and easy to handle, and therefore they are considered to be very effective measurement means in the future. For that reason, we think it is important to use motion analysis equipment, which has become a gold standard, to show the effects of changes in acceleration, as was the case in this research.

## 5. Limitations of This Study

This study had several limitations. First, it targeted patients with bilateral knee osteoarthritis, and contralateral hypofunction may have affected the results of the study. However, in severe KOA, both sides are often affected, and evaluation of the opposite side after TKA should be considered in the future. Second, gait analysis was performed at a relatively early stage (5 weeks after surgery), and sufficient functional recovery could not be ascertained. However, since appropriate rehabilitation can be performed even in the early postoperative period to improve gait, as observed from the results of this study, the obtained information is considered useful for determining the effect of early rehabilitation. Acquiring time-series data not only in the early postoperative period but also at 6 months and 1 year after the surgery is important. In the future, it will be necessary to continuously acquire data over the long term.

## 6. Conclusions

In conclusion, lateral sway of COM in the stance phase of walking after TKA tends to improve in a relatively early stage, and especially in the early and late stance phase, changes in acceleration become smaller and stability in walking is achieved.

## Figures and Tables

**Figure 1 geriatrics-08-00002-f001:**
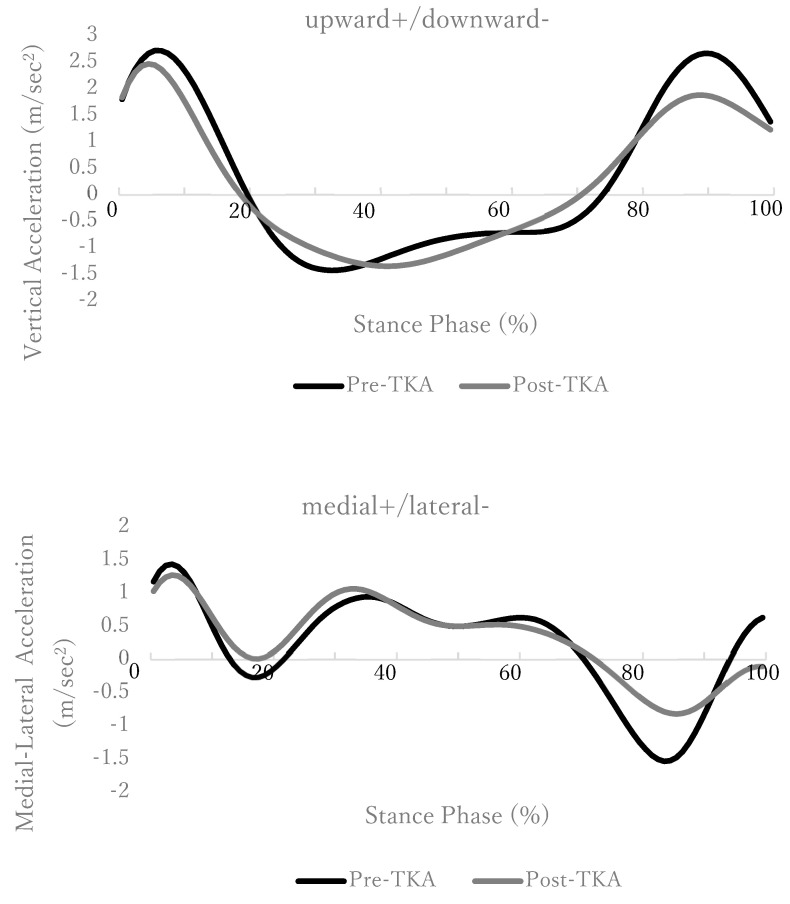
Time-series changes in acceleration during the stance phase of walking. The upper figure shows the vertical acceleration data, and the lower figure is the lateral acceleration data. The black line indicates pre−TKA, and the gray line indicates post−TKA.

**Table 1 geriatrics-08-00002-t001:** Preoperative and postoperative characteristics of the patients.

Variables	Pre-TKA		Post-TKA		*p*-Value	Effect Size (*d*)
	AVE	±SD	AVE	±SD		
Pain score (NRS)	7.83	±1.59	1.58	±1.24	<0.001 *	4.38
ROM in knee extension (°)	−4.58	±8.38	−1.67	±2.46	0.003 *	0.47
ROM in knee flexion (°)	112.50	±18.03	115.25	±13.34	0.080	0.17
FTA (°)	185.58	±5.11	174.83	±2.52	<0.001 *	2.67

* A significant difference is shown when *p* < 0.05. NRS, numerical rating scale; ROM, range of motion; FTA, femorotibial angle.

**Table 2 geriatrics-08-00002-t002:** Pearson correlation coefficients between the RMS and walking speed.

Variables	Walking Speed of Pre-TKA	*p*-Value	Walking Speed of Post-TKA	*p*-Value
	*r*		*r*	
RMS of anterior–posterior direction	−0.28	0.357	−0.61	0.027 *
RMS of lateral direction	−0.33	0.271	−0.18	0.553
RMS of vertical direction	−0.66	0.014 *	−0.63	0.021 *

* A significant difference is shown when *p* < 0.05.

**Table 3 geriatrics-08-00002-t003:** RMS of COM during the walking stance phase.

Variables	Pre-TKA		Post-TKA		*p*-Value	Effect Size (*d*)
	Ave	SD	Ave	SD		
Anterior–posterior direction	1.42	±0.26	1.34	±0.65	0.198	0.16
Lateral direction	0.86	±0.23	0.73	±0.19	0.035 *	0.62
Vertical direction	1.53	±0.39	1.36	±0.36	0.047 *	0.45

* A significant difference is shown when *p* < 0.05.

**Table 4 geriatrics-08-00002-t004:** Changes in lateral acceleration during the walking stance phase.

	Pre−TKA		Post−TKA		*p*-Value	Effect Size (*d*)
	Ave	SD	Ave	SD		
Acceleration in the medial direction at the early stance phase (m/s^2^)	1.51	±0.73	1.35	±0.54	0.327	0.25
Acceleration in the lateral direction at the early stance phase (m/s^2^)	0.58	±0.68	0.10	±0.60	0.007 *	0.75
Acceleration in the lateral direction at the late stance phase (m/s^2^)	1.62	±0.97	0.95	±0.80	<0.001 *	0.75

* A significant difference is shown when *p* < 0.05.

**Table 5 geriatrics-08-00002-t005:** Changes in vertical acceleration during the walking stance phase.

	Pre−TKA		Post−TKA		*p*-Value	Effect Size (*d*)
	Ave	SD	Ave	SD		
Acceleration in the upward direction at the early stance phase (m/s^2^)	2.77	±0.70	2.53	±0.73	0.308	0.34
Acceleration in the downward direction at the stance phase (m/s^2^)	1.85	±0.65	1.81	±0.61	0.818	0.06
Acceleration in the upward direction at the late stance phase (m/s^2^)	3.23	±0.86	2.74	±0.73	0.020 *	0.61

* A significant difference is shown when *p* < 0.05.

## Data Availability

Not applicable.

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
