# Peer review of "Sway and Acceleration Changes of the Center of Mass during Walking Stance Phase before and after Total Knee Arthroplasty"

_geriatrics, 2022, doi:10.3390/geriatrics8010002_

Round 1

Reviewer 1 Report

The authors have done an excellent job, and the introduction is very well-grounded, but still, the proposal is that some changes need to be made:

1.      The proposal is that a minor change to the title be made: article Sway and acceleration changes of the center of mass during walking stance phase before and after total knee arthroplasty.

2.      However, only few reports on the COM sway during walking in KOA are available (lines 51-52).

·        Please kindly insert some citations of the COM reports sway during walking in KOA.

3.      It has been reported to effectively improve pain and function in self-reported assessments, including the Western Ontario and McMaster Universities Osteoarthritis Index [14] (lines 56-58).

·        Please also insert the following bibliographic reference: McConnell S, Kolopack P, Davis AM. The Western Ontario and McMaster Universities Osteoarthritis Index (WOMAC): a review of its utility and measurement properties. Arthritis Rheum. 2001;45(5):453-461. doi:10.1002/1529-0131(200110)45:5<453::aid-art365>3.0.co;2-w

4.      At the end of the introduction, please insert the present research’s novel elements and the paper’s purpose.

5.      Please enter the study's hypothesis in the Materials and Methods section.

6.      All statistical analyses were performed using 155 SPSS software, version 23.0 (SPSS Inc., Tokyo, Japan) (lines 155-156).  Please change with IBM SPSS 23.0 Statistics (IBM Corp. Released 2015. IBM SPSS Statistics for Windows, Version 23.0. IBM Corp., Armonk, NY, USA).

7.      On line 274 please enter 5. Limitations of this study.

8.      The article was included to analyze for the similarity coefficient in the Plagiarism CheckerX software, version 6.0.11, and please perform the following word reformulations that are in bold and italics:

·        According to a lower extremity model of the Plug-In-Gait marker set [20], which is a widely used standardized marker arrangement for three-dimensional motion analysis, 9.5-mm-diameter reflective markers were placed directly over the following bilateral anatomical landmarks: anterior and posterior superior iliac spines, lateral thighs, lateral femoral epicondyles, lateral shanks, lateral malleoli, calcanei, and dorsa of the feet at the base of the second metatarsal.

Author Response

Responses to the comments of Reviewer 1

In the manuscript, stained Red words are revised parts according to your suggestions.

Comment1:

The proposal is that a minor change to the title be made: article Sway and acceleration changes of the center of mass during walking stance phase before and after total knee arthroplasty.

Response of Authors:

In accordance with the Reviewer's comment, we revised as follows.

Sway and acceleration changes of the center of mass during walking stance phase before and after total knee arthroplasty.

Comment2:

However, only few reports on the COM sway during walking in KOA are available (lines 51-52).

Please kindly insert some citations of the COM reports sway during walking in KOA.

Response of Authors:

In accordance with the Reviewer's comment, we have inserted the following documents。

  1. 16. Vahtrik D, Ereline J, Gapeyeva H, Pääsuke M. Postural stability in relation to anthropometric and functional characteristics in women with knee osteoarthritis following total knee arthroplasty. Arch Orthop Trauma Surg. 2014;134(5):685-92. doi: 10.1007/s00402-014-1940-9. Epub 2014 Feb 14.

Comment3:

It has been reported to effectively improve pain and function in self-reported assessments, including the Western Ontario and McMaster Universities Osteoarthritis Index [14] (lines 56-58).

Please also insert the following bibliographic reference: McConnell S, Kolopack P, Davis AM. The Western Ontario and McMaster Universities Osteoarthritis Index (WOMAC): a review of its utility and measurement properties. Arthritis Rheum. 2001;45(5):453-461. doi:10.1002/1529-0131(200110)45:5<453::aid-art365>3.0.co;2-w

Response of Authors:

In accordance with the Reviewer's comment, we have inserted the document below.

[18] McConnell S, Kolopack P, Davis AM. The Western Ontario and McMaster Universities Osteoarthritis Index (WOMAC): a review of its utility and measurement properties. Arthritis Rheum. 2001;45(5):453-461. doi:10.1002/1529-0131(200110)45:5<453::aid-art365>3.0.co;2-w

Comment4:

At the end of the introduction, please insert the present research’s novel elements and the paper’s purpose.

Response of Authors:

In accordance with the Reviewer's comment, we have added the following sentences.

There are few measurements of changes in center of gravity sway due to walking at a relatively early stage after TKA, and we believe that presenting the results is novel.

Comment5:

Please enter the study's hypothesis in the Materials and Methods section.

Response of Authors:

We have already stated the hypothesis in the introduction. Please confirm.

Comment6:

All statistical analyses were performed using 155 SPSS software, version 23.0 (SPSS Inc., Tokyo, Japan) (lines 155-156).  Please change with IBM SPSS 23.0 Statistics (IBM Corp. Released 2015. IBM SPSS Statistics for Windows, Version 23.0. IBM Corp., Armonk, NY, USA).

Response of Authors:

In accordance with the Reviewer's comment, we revised as follows.

IBM SPSS 23.0 Statistics (IBM Corp. Released 2015. IBM SPSS Statistics for Windows, Version 23.0. IBM Corp., Ar-monk, NY, USA).

Comment7:

On line 274 please enter 5. Limitations of this study.

Response of Authors:

In accordance with the Reviewer's comment, we revised as follows.

  1. Limitations of this study

Comment8:

The article was included to analyze for the similarity coefficient in the Plagiarism CheckerX software, version 6.0.11, and please perform the following word reformulations that are in bold and italics:

According to a lower extremity model of the Plug-In-Gait marker set [20], which is a widely used standardized marker arrangement for three-dimensional motion analysis, 9.5-mm-diameter reflective markers were placed directly over the following bilateral anatomical landmarks: anterior and posterior superior iliac spines, lateral thighs, lateral femoral epicondyles, lateral shanks, lateral malleoli, calcanei, and dorsa of the feet at the base of the second metatarsal.

Response of Authors:

In accordance with the Reviewer's comment, we revised as follows.

In this study, three-dimensional motion analysis with a lower extremity model of the Plug-In-Gait marker set [24] was used. Reflective markers of 9.5 mm diameter were affixed to the following anatomic landmarks: anterior and posterior superior iliac spines, lateral thighs, lateral femoral epicondyles, lateral shanks, lateral malleoli, calcanei, and dorsa of the feet at the base of the second metatarsal.

Reviewer 2 Report

Important experiment but the analysis and the presentation of the experiment lack scientific rigour. I suggest that the authors address the following limitations before publishing this work.

Introduction: COM Acceleration

What does acceleration of COM actually measure? Postural control? Dynamic stability? Is COM acceleration the best measure of these phenomena? Is COM acceleration a reliable, accurate and sensitive measure of these phenomena? The authors have used old publications (12, 13) to support the choice of their outcome measure. Please refer to more recent literature to support the choice of the outcome measure. See the following for more detail:

Lord, S.; Galna, B.; Rochester, L., Moving forward on gait measurement: toward a more refined approach. Mov Disord 2013, 28, (11), 1534-43, https://doi.org/10.1002/mds.25545

Abstract

"significant differences between the two groups". Which two groups? Define the two groups before mentioning them. The word "group" is misleading. Probably, the authors mean to say that there were statistically significant differences between pre-operative and post-operative measurements. The study had two "time points" and a single group of people. Not two groups.

Relationship between COM accelerations and walking speed is crucial and should be mentioned in the abstract. The authors should clarify that they measured RMS of COM acceleration normalised by walking speed.

2.4 Data Analysis

The equation for RMS is not adequate. The authors should state the equation for RMS normalised by walking speed. All the symbols used in the equation should also be defined. Not all the readers are familiar with mathematical formulations. Furthermore, a discrete-time formulation should be used instead of the continuous-time formulation. I don't think the authors integrated the acceleration. Rather, they must have applied a discrete-sum which approximated integration.

Acceleration and Walking Speed

I am not convinced that the relationship between acceleration and walking speed can be adequately taken care of by simply normalising acceleration by walking speed. Please convince me otherwise by presenting a scatter plot of normalised COM acceleration vs walking speed.

Also report a statistical analysis for the difference in walking speed across pre-operative and post-operative time points. I suspect the COM acceleration is acting as a proxy of the following latent phenomena: pace, postural control, etc. See following for more details:

Lord, S.; Galna, B.; Rochester, L., Moving forward on gait measurement: toward a more refined approach. Mov Disord 2013, 28, (11), 1534-43, https://doi.org/10.1002/mds.25545

2.5 Statistical Analysis

Was the data normally distributed? t-test assumes normality. Please provide a qq-plot for each outcome measure.

How was the early-stance phase and late-stance phase defined? 

3 Results

First present the results of early stance phase and then the late stance phase. What is the difference between table 2 and 3?

Clinical Implications

Please discuss the clinical implications of the findings in reference to emerging technologies such as gait measurement with mobile devices. Can clinicians measure the COM acceleration of their patients in a clinic using a smartphone to personalise physical therapy of their patients? For details see:

Rashid, Usman, et al. "Validity and reliability of a smartphone app for gait and balance assessment." Sensors 22.1 (2021): 124.

Conclusion

The conclusion is incorrect. Please point out the evidence which supports the causal statement that "this study showed that the improvement in pain and alignment after TKA reduced the lateral sway of COM". Is Y caused by X or simply correlated with X? For details, see the following work and reword the conclusion:

Russo, F. (2015). Causation and Correlation in Medical Science: Theoretical Problems. In T. Schramme, & S. Edwards (Eds.), Handbook of the Philosophy of Medicine (pp. 1-11). Springer Netherlands. http://link.springer.com/referenceworkentry/10.1007/978-94-017-8706-2_46-1

Author Response

Responses to the comments of Reviewer 2

In the manuscript, stained Blue words are revised parts according to your suggestions.

Comment1:

Introduction: COM Acceleration

What does acceleration of COM actually measure? Postural control? Dynamic stability? Is COM acceleration the best measure of these phenomena? Is COM acceleration a reliable, accurate and sensitive measure of these phenomena? The authors have used old publications (12, 13) to support the choice of their outcome measure. Please refer to more recent literature to support the choice of the outcome measure. See the following for more detail:

Response of Authors:

In accordance with the Reviewer's comment, we added the following previous research on the usefulness of using acceleration as a measurement tool for the dynamic stability of walking.

COM acceleration contributes to walking stability [14] and a decrease in mediolateral COM accelerations alters gait patterns in the elderly [15].

Comment2:

Abstract

"significant differences between the two groups". Which two groups? Define the two groups before mentioning them. The word "group" is misleading. Probably, the authors mean to say that there were statistically significant differences between pre-operative and post-operative measurements. The study had two "time points" and a single group of people. Not two groups.

Response of Authors:

In accordance with the Reviewer's comment, we have corrected the description of the two groups in the summary by dividing them into pre-operative and post-operative.

Lateral and vertical RMS showed significant differences between pre-operative and post-operative, and demonstrated low values after TKA. The lateral acceleration at the latter part of the early stance phase demonstrated a significant difference between pre-operative and post-operative.

Comment3:

Relationship between COM accelerations and walking speed is crucial and should be mentioned in the abstract. The authors should clarify that they measured RMS of COM acceleration normalised by walking speed.

Response of Authors:

In accordance with the Reviewer's comment, we added the following sentences to the abstract.

The COM sway during the stance phase of walking was evaluated using root mean square (RMS) normalized by walking speed, and the changes in acceleration were further verified.

Comment4:

2.4 Data Analysis

The equation for RMS is not adequate. The authors should state the equation for RMS normalised by walking speed. All the symbols used in the equation should also be defined. Not all the readers are familiar with mathematical formulations. Furthermore, a discrete-time formulation should be used instead of the continuous-time formulation. I don't think the authors integrated the acceleration. Rather, they must have applied a discrete-sum which approximated integration.

Response of Authors:

In accordance with the Reviewer's comment, we have modified the RMS formula and added definitions for the symbols as follows.

RMS=

In the above formula, α indicates acceleration, and n indicates the number of frames of measurement data.

Comment5:

Acceleration and Walking Speed

I am not convinced that the relationship between acceleration and walking speed can be adequately taken care of by simply normalising acceleration by walking speed. Please convince me otherwise by presenting a scatter plot of normalised COM acceleration vs walking speed.

Response of Authors:

We standardized the acceleration results of this study with reference to the following paper, which reports that acceleration is exponentially proportional to walking speed based on the relationship between walking speed and acceleration. In addition, the following documents have been added to the references.

Menz HB, Lord SR, Fitzpatrick RC. Age-related differences in walking stability. Age Ageing. 2003. ;32(2):137-42. doi: 10.1093/ageing/32.2.137.

Comment6:

Also report a statistical analysis for the difference in walking speed across pre-operative and post-operative time points. I suspect the COM acceleration is acting as a proxy of the following latent phenomena: pace, postural control, etc. See following for more details:

Response of Authors:

In accordance with the Reviewer's comment, we added walking speed statistics.

3.1. Walking speed and Sway of the COM between groups

In this study, there was no significant difference in walking speed between pre and post TKA (p=.22).

Comment7:

2.5 Statistical Analysis

Was the data normally distributed? t-test assumes normality. Please provide a qq-plot for each outcome measure.

Response of Authors:

We performed normality tests on all variables. As a result, the qq plot showed linearity, confirming normality. Therefore, we performed statistical analysis by t-test.

Comment8:

How was the early-stance phase and late-stance phase defined?

Response of Authors:

In accordance with the Reviewer's comment, we added the following sentences.

In this study, early stance shows data up to 20% of the stance phase, and late stance shows data after 80% of the stance phase.

Comment9:

3 Results

First present the results of early stance phase and then the late stance phase. What is the difference between table 2 and 3?

Response of Authors:

Table 2 shows changes in RMS during the stance phase of walking, and Tables 3 and 4 show characteristic acceleration changes in the early and late stages of walking.

Comment10:

Clinical Implications

Please discuss the clinical implications of the findings in reference to emerging technologies such as gait measurement with mobile devices. Can clinicians measure the COM acceleration of their patients in a clinic using a smartphone to personalise physical therapy of their patients? For details see:

Response of Authors:

In accordance with the Reviewer's comment, we added the following sentences in Discussion and added more literature.

Recently, analysis using smartphones and general cameras has been reported [33]. Although there are issues such as measurement accuracy that can withstand the validity of clinical measurement results, these methods are clinically simple and easy to handle, so they are considered to be very effective measurement means in the future. For that reason, we think it is important to use motion analysis equipment, which has become a gold standard, to show the effects of changes in acceleration, as in this research.

33.Rashid U, Barbado D, Olsen S, Alder G, Elvira JL, Lord S, Niazi IK, Taylor D. Validity and Reliability of a Smartphone App for Gait and Balance Assessment. Sensors (Basel). 2021 ;22(1):124. doi: 10.3390/s22010124.

Comment11:

Conclusion

The conclusion is incorrect. Please point out the evidence which supports the causal statement that "this study showed that the improvement in pain and alignment after TKA reduced the lateral sway of COM". Is Y caused by X or simply correlated with X? For details, see the following work and reword the conclusion:

Response of Authors:

In accordance with the Reviewer's comment, we changed it as below.

In conclusion, Lateral sway of COM in the stance phase of walking after TKA tends to improve in a relatively early stage, and especially in the early and late stance phase, changes in accel-eration become smaller and stability in walking is achieved.

Round 2

Reviewer 1 Report

Congratulations for your hard work in producing this scientific material!

Author Response

Thank you for reviewing my paper. We was able to revise the manuscript to a better one with the appropriate comments from the Reviewer.

Reviewer 2 Report

Thank you for your response. I am not satisfied with the response to my following comment from the previous round.

"I am not convinced that the relationship between acceleration and walking speed can be adequately taken care of by simply normalising acceleration by walking speed. Please convince me otherwise by presenting a scatter plot of normalised COM acceleration vs walking speed."

I am aware of the paper that you have cited. Please present your own data. A scatter plot should be presented between normalised COM acceleration and walking speed so that the readers can contextualise your results.

Author Response

We submitted the manuscript (Manuscript ID: geriatrics-1979703). We then reconsidered the manuscript and revised it according to the reviewers' suggestions. Therefore, we would like to request a re-review of the manuscript.

Suggestions of the reviewers were highly insightful and enabled us to greatly improve the quality of our manuscript. In the following, we wrote our point-by-point responses to each of the comments of the reviewer.

We shall look forward to hearing from you at your earliest convenience.

Yours sincerely,
